# Exploiting Connections between Lipschitz Structures for Certifiably Robust Deep Equilibrium Models

Aaron J. Havens[*1]    Alexandre Araujo[*2]    Siddharth Garg[2]
Farshad Khorrami[2]    Bin Hu[1]

[1] CSL & ECE, University of Illinois Urbana-Champaign
[2] ECE, New York University

## Abstract

Recently, deep equilibrium models (DEQs) have drawn increasing attention from the machine learning community. However, DEQs are much less understood in terms of certified robustness than their explicit network counterparts. In this paper, we advance the understanding of certified robustness of DEQs via exploiting the connections between various Lipschitz network parameterizations for both explicit and implicit models. Importantly, we show that various popular Lipschitz network structures, including convex potential layers (CPL), SDP-based Lipschitz layers (SLL), almost orthogonal layers (AOL), Sandwich layers, and monotone DEQs (MonDEQ) can all be reparameterized as special cases of the Lipschitz-bounded equilibrium networks (LBEN) without changing the prescribed Lipschitz constant in the original network parameterization. A key feature of our reparameterization technique is that it preserves the Lipschitz prescription used in different structures. This opens the possibility of achieving improved certified robustness of DEQs via a combination of network reparameterization, structure-preserving regularization, and LBEN-based finetuning. We also support our theoretical understanding with new empirical results, which show that our proposed method improves the certified robust accuracy of DEQs on classification tasks. All codes and experiments are made available at https://github.com/AaronHavens/ExploitingLipschitzDEQ.

## 1 Introduction

Recently, deep equilibrium models (DEQs) have drawn increasing attention from the deep learning community [3]. In DEQs, the output is defined as the solution to an input-dependent fixed point equation. Specifically, consider a DEQ with input $x$ and output $y$. Then $y$ is typically determined from $x$ as follows

$$z = \sigma(Wz + Ux + b_z), \quad y = Gz + b_y, \tag{1}$$

where $\sigma$ denotes some nonlinear activation function, e.g. ReLU. DEQs can be viewed as an implicit model that directly solves the fixed point of an infinitely-deep network. Since DEQs have implicit depth, they have the potential to reduce the large memory footprint required to train finite-depth feed-forward networks via explicit back-propagation. Noticeably, DEQs have shown great promise in achieving comparable performance on computer vision tasks [4, 33, 6, 28, 23].

Despite great potential, there are many open issues to be explored and understood for DEQs. There has been a body of recent works focusing on developing and understanding DEQ structures for addressing well-posedness, training stability, and adversarial robustness [47, 35, 31, 7, 30, 46, 5, 17, 25, 48, 13].

---

[*] Equal contribution.

37th Conference on Neural Information Processing Systems (NeurIPS 2023).

For example, the monotone DEQ (MonDEQ) uses the following parameterization:

$$z = \sigma\left(((1-m)I - A^\top A + B - B^\top)z + Ux + b_z\right), \quad y = Gz + b_y,$$

where $(A, B, U, G, b_z, b_y)$ are the decision variables to be trained, and $m$ is a hyperparameter [47]. In other words, MonDEQ does not treat $W$ as the variable to be directly trained and just parameterizes it as $W = (1-m)I - A^\top A + B - B^\top$. The advantage of this parameterization is that well-posedness and $\frac{\|U\|_2}{m}$-Lipschitzness (from $x$ to $z$) are automatically ensured [47, 31]. Another popular DEQ structure for controlling the Lipschitz constant is given by the so-called Lipschitz-bounded equilibrium network (LBEN) parameterization which specifies the weight matrix $W$ in (1) as follows

$$W = I - \Psi((G^\top G + \Psi^{-1}UU^\top\Psi^{-1})/(2L) + V^\top V + S - S^\top + \epsilon I), \tag{2}$$

where $\Psi$ is a diagonal positive definite matrix, and $(V, S)$ are unconstrained free variables [35]. Using the LBEN parameterization, one can train $L$-Lipschitz DEQs by solving an unconstrained optimization problem.

This paper focuses on advancing the understanding of DEQ parameterizations in the context of certified robustness. It is well known that neural networks are susceptible to imperceptible adversarial input perturbations [40]. Hence, there is a need to develop certifiably robust models for safety-critical applications. For explicit networks, there has been significant progress on improving certified robustness by using various 1-Lipschitz layer structures such as orthogonal layers [41, 24, 37, 49], almost orthogonal layers (AOL) [34], convex potential layers (CPL) [27], SDP-based Lipschitz layers (SSL) [2], and Sandwich layers [43]. However, how to train DEQs with reasonable certified robustness remains largely open [31]. In this paper, we improve the certified robustness of DEQs via exploiting the connections between various Lipschitz structures for both explicit and implicit models. Importantly, we show that various popular Lipschitz network structures, including convex potential layers (CPL), SDP-based Lipschitz layers (SLL), almost orthogonal layers (AOL), Sandwich layers, and monotone DEQs (MonDEQ) can all be reparameterized as special cases of the Lipschitz-bounded equilibrium networks (LBEN) without changing the prescribed Lipschitz constant in the original network parameterizations. A key feature of our reparameterization technique is that it preserves the Lipschitz prescription used in different structures. This feature is particular relevant to training certifiably robust DEQs, and opens the possibility of achieving improved certified robustness of DEQs via a combination of network reparameterization, structure-preserving regularization, and LBEN-based fine-tuning. We provide empirical results to show that our reparameterization technique improves the certified robust accuracy of DEQs on classification benchmarks. Our work sheds new light on the principles for developing 1-Lipschitz DEQ structures and serves as a meaningful step towards achieving better certified robustness for DEQs.

**Additional Related Work.** There are also several recent results on certified robustness of DEQs under $\ell_\infty$ perturbations [22, 30, 46, 11, 18, 10]. Our work focuses on certified robustness under $\ell_2$ perturbations and hence naturally complements these existing works. In this paper, we mainly study the impacts of network parameterizations on certified robustness of DEQs. For explicit models, many other methods for designing certifiably robust models are also available [8, 36, 21, 16, 38, 15]. It is an interesting future task to study how to tailor these different methods for improving the certified robust accuracy of DEQs.

## 2 Background and Preliminaries

**Notation.** The set of $n$-dimensional real vectors is denoted by $\mathbb{R}^n$. For any vector $x \in \mathbb{R}^n$, we denote the associated Euclidean norm as $\|x\|$. In addition, we denote the $n \times n$ identity matrix and the $n \times n$ zero matrix as $I_n$ and $0_n$, respectively. The subscripts will be omitted when the dimension is clear from the context. When a matrix $P$ is negative semidefinite (definite), we will use the notation $P \preceq (\prec)0$. When a matrix $P$ is positive semidefinite (definite), we will use the notation $P \succeq (\succ)0$. Given a collection of scalars $\{a_i\}_{i=1}^n$, the $n \times n$ diagonal matrix whose $(i, i)$-th entry is $a_i$ is denoted as $\mathrm{diag}(a_i)$. For a matrix $A$, let $A^\top$ and $\|A\|_2$ denote its transpose and largest singular value, respectively.

### 2.1 Lipschitz Property and Certified Robustness

Suppose we have a prediction model given by an input-output mapping $y = f(x)$. We say that $f$ is $L$-Lipschitz (with respect to the $\ell_2$ norm) if $\|f(x_1) - f(x_2)\| \leq L\|x_1 - x_2\|$ for any $(x_1, x_2)$.

The Lipschitz constant can be combined with the prediction margin of $f$ to give certified robust accuracy [42, Proposition 1]. In the past, 1-Lipschitz models have shown great promise in achieving improved certified robustness on classification benchmark problems[29, 41, 24, 37, 27, 34, 2]. In this paper, we focus on designing certifiably robust DEQs via using novel 1-Lipschitz parameterizations.

## 2.2 SDPs for Well-posedness and Lipschitz Bounds of DEQs

Given a DEQ in the general form (1), the following existing sufficient conditions are useful for testing its well-posedness and Lipschitz properties.

**Proposition 2.1** (Theorem 1 in Revay et al. [35]). *Consider the DEQ model* (1) *with $\sigma$ being sloped-restricted on $[0, 1]$. If there exists a positive definite diagonal matrix $\Gamma$ such that the following SDP holds,*

$$2\Gamma - \Gamma W - W^\top \Gamma \succ 0, \tag{3}$$

*then the DEQ model* (1) *is well-posed and has a finite Lipschitz bound from $x$ to $y$.*

**Proposition 2.2** (Theorem 2 in Revay et al. [35]). *Consider the DEQ model* (1) *with $\sigma$ be slope-restricted on $[0, 1]$. If there exists a positive definite diagonal matrix $\Lambda$ such that*

$$2\Lambda - \Lambda W - W^\top \Lambda - \frac{1}{L}G^\top G - \frac{1}{L}\Lambda U U^\top \Lambda \succ 0, \tag{4}$$

*then the DEQ* (1) *is well-posed and $L$-Lipschitz from $x$ to $y$.*

The above propositions can be proved using standard control-theoretic arguments borrowed from the quadratic constraint theory [26, 12]. See [35] for detailed proofs. In general, matrix inequality conditions have been widely adopted for addressing the Lipschitz properties of neural networks [12, 32, 14, 44, 2, 45, 43], and (4) can be viewed as the extension of the LipSDP condition [12] to the DEQ setting. Notice that we use different notations for the decision variables in the testing conditions (3) and (4) to emphasize the fact that the solution to (3) may not be a solution to (4) even for the same DEQ model. For fixed $W$, the condition (3) is an SDP that can be efficiently verified by existing solvers. For given $(W, U, G)$, the condition (4) is bilinear in $\Lambda$ and hence not an SDP. However, one can easily convexify (4) into an SDP form using the Schur complement lemma. Later, we will develop a variant of Proposition 2.2, which can be used to unify the developments of various Lipschitz implicit and explicit models.

## 2.3 DEQ Parameterizations: MonDEQ and LBEN

Now we briefly review several existing DEQ parameterizations for inducing well-posedness and the Lipschitz property.

**MonDEQ.** The well-posedness issue of DEQs was addressed in [47] using the MonDEQ parameterization derived from the monotone operator theory. In addition, operator splitting approaches can be used to efficiently compute the unique fixed point of MonDEQ. Recall that MonDEQ adopts the following weight parameterization:

$$W = (1 - m)I - A^\top A + B - B^\top \tag{5}$$

where $m > 0$, and $(A, B)$ are free decision variables to be trained. As long as $W$ is parameterized as in (5), the resultant DEQ model (1) is well-posed [47]. Based on examining the unrolled operator splitting iterates, it has been shown [31] that MonDEQ is $\frac{\|U\|_2}{m}$-Lipschitz from $x$ to $z$ (and hence $\frac{\|U\|_2 \|G\|_2}{m}$-Lipschitz from $x$ to $y$). If $G = I$, then one can enforce MonDEQ to be 1-Lipschitz by adding the constraint $\|U\|_2 \leq m$.

**Generalized MonDEQ (G-MonDEQ).** Revay et al. [35] generalizes the MonDEQ parameterization as follows

$$W = I - \Phi(A^\top A + B^\top - B + mI), \tag{6}$$

where $\Phi$ is a positive definite diagonal matrix to be trained. Obviously, the MonDEQ model is a special case of (6) with $\Phi = I$. For the G-MonDEQ parameterization (6), the well-posedness can also be guaranteed. Specifically, one can verify that the SDP condition (3) with $W$ given by (6) holds if we choose $\Gamma = \Phi^{-1}$. In this case, the left side of (3) becomes $2A^\top A + 2mI \succ 0$. The Lipschitz constant for the G-MonDEQ parameterization (6) has not been derived before. Later we will present an explicit Lipschitz bound for (6) via constructing an analytical solution to a variant of (4).

**LBEN.** Revay et al. [35] has also proposed the following LBEN parameterization with free decision variables $(G, U, V, S)$, a diagonal matrix variable $\Psi \succ 0$, and a hyperparameter $\varepsilon > 0$:

$$W = I - \Psi\left(\frac{1}{2L}G^\top G + \frac{1}{2L}\Psi^{-1}UU^\top\Psi^{-1} + V^\top V + S^\top - S + \varepsilon I\right). \tag{7}$$

The LBEN parameterization guarantees the DEQ to be well-posed and $L$-Lipschitz. Specifically, one can verify that the condition (4) holds with $W$ being defined by (7) and $\Lambda = \Psi^{-1}$. In this case, the left side of (4) becomes $2V^\top V + 2\varepsilon I \succ 0$. The above parameterization allows one to train $L$-Lipschitz DEQs in an unconstrained manner.

### 2.4 1-Lipschitz Explicit Feed-forward Networks: AOL, SLL and Sandwich Layers

For traditional explicit models, there exist several 1-Lipschitz parameterizations which can be used for efficient training of certifiably robust feed-forward networks. Now we briefly discuss several state-of-the-art 1-Lipschitz feed-forward parameterizations.

**Orthogonal and Almost Orthogonal Layers.** Consider the following standard explicit feed-forward network:

$$x_0 = x, \quad x_{n+1} = \sigma(W_n x_n + b_n), \quad y = W_N x_N + b_N,$$

where $n = 0, 1, \ldots, N - 1$. Since the compositions of 1-Lipschitz functions are also 1-Lipschitz, one can just parameterize every explicit layer $x_{n+1} = \sigma(W_n x_n + b_n)$ to be 1-Lipschitz, and the resultant feed-forward network is 1-Lipschitz. If $\sigma$ is 1-Lipschitz[1], one only needs to ensure $\|W_n\|_2 \leq 1$ for all $n$. An important issue in training such Lipschitz layers for deep explicit networks is gradient vanishing, and this motivates the development of various techniques which introduce gradient norm preservation based on orthogonality. The orthogonality parameterizations just enforce $W_n$ to satisfy $W_n^\top W_n = I$ [41, 24, 37]. In contrast, the *Almost-Orthogonal-layer* (AOL) [34] performs a weight normalization $W_n = \tilde{W}_n D_n$, where $\tilde{W}_n$ is the free decision variable to be trained, and $D_n$ is a scaling matrix which is diagonal and defined as[2]

$$D_n = \text{diag}\left(\sum_j |\tilde{W}_n^\top \tilde{W}_n|_{ij}\right)^{-\frac{1}{2}}. \tag{8}$$

AOL ensures $\|W_n\|_2 \leq 1$. In [34], it has been empirically demonstrated that the trained weight $W_n$ tends to be "almost orthogonal."

**Lipschitz Residual Networks and SLL.** Another way to address the gradient vanishing issue in training Lipschitz networks is to use a residual structure [27, 2]. A general result is given in [2] and shows that the following residual network is guaranteed to be 1-Lipschitz given $W_n^\top W_n \preceq T_n$:

$$x_0 = x, \quad x_{n+1} = x_n - 2W_n T_n^{-1}\sigma\left(W_n^\top x_n + b_n\right), \quad y = x_N.$$

For CPL, the choice of $T_n$ is simple, i.e. $T_n = \|W_n\|_2^2 I$. To achieve better certified robust accuracy, SLL [2] uses the following more delicate choice of $T_n$:

$$T_n = \text{diag}\left(\sum_j \left|W_n^\top W_n\right|_{ij} \frac{q_j^{(n)}}{q_i^{(n)}}\right) \tag{9}$$

where $\{q_i^{(n)}\}$ are free positive scalars to be trained.

**Sandwich Layers.** Recently, the sandwich layer has been proposed in [43], and achieves very competitive performances with relatively small models. Based on [43], the following network is guaranteed to be $L$-Lipschitz:

$$x_0 = \sqrt{L}x, \quad x_{n+1} = \sqrt{2}A_n^\top \Phi_n \sigma(\sqrt{2}\Phi_n^{-1}B_n x_n + b_n), \quad y = \sqrt{L}B_N x_N + b_N,$$

where $A_n A_n^\top + B_n B_n^\top = I$, $B_N^\top B_N = I$, and $\Phi_n$ is a free diagonal matrix whose entries are restricted to be non-negative. One can parameterize $[A_n \quad B_n]^\top$ to satisfy $A_n A_n^\top + B_n B_n^\top = I$ using the generalized Cayley transformation [19], and train the above network efficiently.

---

[1]Almost all the activation functions that are commonly used in practice are 1-Lipschitz.
[2]If one has $\sum_j |\tilde{W}_n^\top \tilde{W}_n|_{ij} = 0$ for $i$, then the $(i, i)$-th entry of $D_n$ is just set to be 0.

# 3 Network Reparameterizations with Preserved Lipschitz Prescriptions

In this section, we show that various Lipschitz structures such as MonDEQ, G-MonDEQ, SLL, Sandwich, and AOL can all be reparameterized as special cases of LBEN without changing the prescribed Lipschitz constant in the original parameterizations. The feature of preserving the Lipschitz constant prescription between different parameterizations is particularly relevant to training certifiably robust DEQs, and we will elaborate on this point later. We emphasize that our result does not indicate that we should abandon all other Lipschitz structures and only use the LBEN parameterization for training certifiably robust DEQs. As a matter of fact, when implementing these different parameterizations for classification tasks, one can potentially incorporate quite different convolution structures and the inductive bias can be very different. The true implication of our theory is that one can potentially fine-tune the trained models from MonDEQ, SLL, or other explicit Lipschitz layers by reparameterizing the networks as LBEN and initializing the LBEN training from these reparameterized models. The key to this implication is that the prescribed Lipschitz constant has to be preserved during the reparameterization process. Our theory opens the possibility of achieving improved certified robustness of DEQs via a combination of network reparameterization and LBEN-based fine-tuning.

## 3.1 Warm-up: Issues in Reparameterizing MonDEQ as LBEN and a Fix

In this section, we reveal a deeper connection between MonDEQ and LBEN. As mentioned in [35, Remark 2], any MonDEQ immediately leads to an LBEN for sufficiently large $L$. However, this trivial reparameterization of LBEN from MonDEQ does not preserve the prescribed Lipschitz constant in the original MonDEQ parameterization. Specifically, the original MonDEQ controls its Lipschitz constant by $\frac{\|U\|_2}{m}$. After the trivial reparameterization based on [35, Remark 2], the resultant Lipschtiz constant is no longer $\frac{\|U\|_2}{m}$. In contrast, it can be arbitrarily large. This is problematic for training certifiably robust DEQs, since the increase in the Lipschitz constant will decrease the certified robust accuracy.

Notice that the construction of LBEN relies on Proposition 2.2. The above technical issue hinges upon the fact that Proposition 2.2 cannot be directly applied to prove the Lipschitz constant of MonDEQ. To gain some insight into how to fix this issue, we first discuss how to modify Proposition 2.2 to recover the previous Lipschitz bound of MonDEQ. Specifically, we will use a variant of [35, Theorem 2] (which has been restated as Proposition 2.2). Now we state this new condition.

**Theorem 3.1.** *Suppose the DEQ model* (1) *is well-posed, and* $\sigma$ *is slope-restricted on* $[0, 1]$. *If there exists a positive definite diagonal matrix* $\Lambda$ *such that the following non-strict matrix inequality holds,*

$$2\Lambda - \Lambda W - W^\top \Lambda - \frac{1}{L} G^\top G - \frac{1}{L} \Lambda U U^\top \Lambda \succeq 0, \qquad (10)$$

*then the DEQ* (1) *is* $L$-*Lipschitz from* $x$ *to* $y$.

*Proof.* We only need to modify the quadratic constraint argument used in [35, Theorem 2]. Originally, a strict matrix inequality is needed to ensure well-posedness. However, well-posedness has been assumed here, and a non-strict matrix inequality is sufficient to ensure the $L$-Lipschitz property. For completeness, a detailed proof is presented in the appendix. $\qquad \square$

The above variant of the strict matrix inequality (4) gives us a non-strict condition, which can then be used to derive Lipschitz properties of MonDEQ, G-MonDEQ, and other explicit models in a unified manner. It turns out that such a non-strict variant is quite crucial for our proposed reparameterization techniques. Recall that originally the LBEN parameterization (7) is derived using Proposition 2.2. Since Theorem 3.1 is the non-strict variant of Proposition 2.2, it is not that surprising that we can use Proposition 2.1 to take care of the well-posedness issue and then combine it with Theorem 3.1 to prove the desired Lipschitz property. We will show that the Lipschitz properties of MonDEQ, G-MonDEQ, and explicit layers such as SLL can also be obtained by providing analytical solutions to (10). More importantly, these new analytical solutions will guide us to find a systematic way to reparameterize these structures as LBEN without changing the prescribed Lipschitz constant in the original parameterizations.

First, we will apply our new theorem to recover the original Lipschitz bound from [31]. The original analysis in [31] relies on examining the operator splitting iterates and bounding the difference between outputs based on their inputs. Alternatively, we can recover this result using the non-strict condition (10). Specifically, we can set $G = I$ (recalling that we are interested in the Lipschitz constant from $x$ to $z$), $\Lambda = \frac{1}{\|U\|_2}I$, $L = \frac{\|U\|_2}{m}$, and $W$ as defined by the MonDEQ parameterization (5). Then we can show that the left side of (10) becomes

$$\frac{2m}{\|U\|_2}I + \frac{2}{\|U\|_2}A^\top A - \frac{1}{L}I - \frac{1}{L\|U\|^2}UU^\top = \frac{1}{L}I + \frac{2}{\|U\|_2}A^\top A - \frac{1}{L\|U\|_2^2}UU^\top$$

$$= \frac{2}{\|U\|_2}A^\top A + \frac{1}{L}\left(I - \frac{1}{\|U\|_2^2}UU^\top\right) \succeq 0.$$

Therefore, the condition holds with $\Lambda = \frac{1}{\|U\|_2}I$ and $L = \|U\|_2/m$. Notice that MonDEQ is known to be well-posed. Hence Theorem 3.1 can be directly applied to give the conclusion that MonDEQ is $\frac{\|U\|_2}{m}$-Lipschitz from $x$ to $z$. If we have $G = I$ and want to enforce MonDEQ to be 1-Lipschitz, then we only need to require $\|U\|_2 \leq m$. There are many ways to parameterize $U$ such that this norm constraint is automatically satisfied. For example, one can enforce $U/\sqrt{m}$ to be an orthogonal matrix. Similarly, one can also use the scaling trick in AOL to ensure $\|U\|_2 \leq m$. See the appendix for more discussions on how to parameterize 1-Lipschitz MonDEQ.

Next, we discuss how to address the previous issue in reparameterization such that we can reparameterize 1-Lipschitz MonDEQ as LBEN by exactly setting $L = 1$ in the LBEN reparameterization. We will leverage the construction of $\Lambda$ above. Recall that the free variables in MonDEQ are $(A, B, U)$. The matrices $(W, G)$ are given by

$$W = (1 - m)I - A^\top A + B - B^\top, \ G = I$$

Now we confine our scope to 1-Lipschitz MonDEQ, i.e. $\|U\|_2 \leq m$. We will show that given a MonDEQ model (5) with any fixed $(A, B, U)$, we can always generate an LBEN parameterization in the form of (7) which gives exactly the same input-output mapping behavior from $x$ to $y$. This means that any trained MonDEQ model can be reparameterized in the form of LBEN with some choice of $(\Psi, G, U, V, S, \epsilon)$. We will provide explicit formulas for such a choice of $(\Psi, G, U, V, S, \epsilon)$. We have the following formal statement.

**Theorem 3.2.** *Given a 1-Lipschitz MonDEQ model with $(A, B, U)$ satisfying $\|U\|_2 \leq m$, the following choice of $(\Psi, G, U, V, S, \epsilon)$ with $U$ being the same one used for MonDEQ gives a valid 1-Lipschitz LBEN model which generates the same input-output mapping:*

$$\Psi = \|U\|_2 I, \quad G = I, \quad S = \frac{B}{\|U\|_2}, \quad \epsilon = 0$$

$$V^\top V = \left(\frac{m}{\|U\|_2} - \frac{1}{2}\right)I + \frac{A^\top A}{\|U\|_2} - \frac{1}{2\|U\|_2^2}UU^\top \succeq 0$$

*Proof.* The above formula can be easily verified by substituting the expressions for $W$ into both the MonDEQ and LBEN parameterizations and then making simplifications. The only thing that needs to be further checked is that the resultant expression for $V^\top V$ must be positive semidefinite so we can obtain $V$ by decomposition. Since we know $\|U\|_2 \leq m$, we must have

$$V^\top V = \left(\frac{m}{\|U\|_2} - \frac{1}{2}\right)I + \frac{A^\top A}{\|U\|_2} - \frac{1}{2\|U\|_2^2}UU^\top \tag{11}$$

$$\succeq \frac{A^\top A}{\|U\|_2} + \frac{1}{2}\left(I - \frac{1}{\|U\|_2^2}UU^\top\right) \succeq 0 \tag{12}$$

Hence the resultant expression for $V^\top V$ is positive semidefinite. This completes the proof. □

We can clearly see that in the above result, the Lipschitz constant is preserved during the reparameterization.

## 3.2 Connections between G-MonDEQ and LBEN

The Lipschitz constant of the G-MonDEQ parameterization (6) has not been explicitly provided by the original work in [35]. Now we will present such an explicit formula via applying (10). Formally, we have the following result.

**Theorem 3.3.** *Consider the G-MonDEQ parameterization* (6). *It is guaranteed that* (6) *gives a mapping which is* $\frac{\|\Phi^{-1}U\|}{m}$*-Lipschitz from $x$ to $z$.*

*Proof.* We are interested in the Lipschitz constant from $x$ to $z$, and hence we choose $G = I$. As explained before, the G-MonDEQ parameterization is known to be well-posed based on Proposition 2.1. Hence we can apply Theorem 3.1 to upper bound its Lipschitz constant. Specifically, we can choose $\Lambda = \frac{1}{\|\Phi^{-1}U\|_2}\Phi^{-1}$, and $L = \frac{\|\Phi^{-1}U\|_2}{m}$. Then the left side of (10) becomes

$$\frac{1}{L}I + \frac{2}{\|\Phi^{-1}U\|_2}A^\top A - \frac{1}{L\|\Phi^{-1}U\|_2^2}\Phi^{-1}UU^\top\Phi^{-1}$$

$$= \frac{2}{\|\Phi^{-1}U\|_2}A^\top A + \frac{1}{L}\left(I - \frac{1}{\|\Phi^{-1}U\|_2^2}\Phi^{-1}UU^\top\Phi^{-1}\right),$$

which is obviously positive semidefinite. Therefore, (10) is feasible for G-MonDEQ with $L = \frac{\|\Phi^{-1}U\|_2}{m}$. This leads to the desired conclusion. $\qquad\square$

If we choose $\Phi = I$, the above theorem just recovers [31, Theorem 1] as a special case. From the above theorem, we can ensure G-MonDEQ to be 1-Lipschitz by enforcing $\|\Phi^{-1}U\|_2 \le m$ during training.

Next, we can leverage the above construction of $\Lambda$ to establish a similar reparameterization result connecting G-MonDEQ and LBEN.

**Theorem 3.4.** *Given a 1-Lipschitz G-MonDEQ model with $(\Phi, A, B, U)$ satisfying $\|\Phi^{-1}U\|_2 \le m$, the following choice of $(\Psi, G, U, V, S, \epsilon)$ with $U$ being the same one used for G-MonDEQ gives a valid 1-Lipschitz LBEN model which generates exactly the same input-output mapping:*

$$\Psi = \|\Phi^{-1}U\|_2\Phi, \quad G = I, \quad S = \frac{1}{\|\Phi^{-1}U\|}B, \quad \epsilon = 0$$

$$V^\top V = \left(\frac{m}{\|\Phi^{-1}U\|_2} - \frac{1}{2}\right)I + \frac{A^\top A}{\|\Phi^{-1}U\|_2} - \frac{1}{2\|\Phi^{-1}U\|_2^2}\Phi^{-1}UU^\top\Phi^{-1} \succeq 0.$$

*Proof.* The proof is very similar to the proof of the MonDEQ case and hence omitted. $\qquad\square$

Again, the prescribed Lipschitz constant is preserved during our reparameterization.

## 3.3 Connections between SLL and LBEN

Extending the previous analysis, we can further build connections between LBEN and explicit networks such as SLL. The SLL network is parameterized by $\{W_n, T_n\}$ with $W_n^\top W_n \preceq T_n$. Again, we can choose $(\Psi, G, U, V, S, \epsilon)$ properly to generate a valid 1-Lipschitz LBEN which gives the same input-output mapping. Similar to the previous treatment, we will first recover the Lipschitz analysis of SLL to gain some insights on how to construct $\Lambda$ for the reparameterization. Notice that the feed-forward network is always well-posed. Hence we can just rewrite a feed-forward residual network in the form of (1) and then apply Theorem 3.1 to analyze its Lipschitz property. To see this, consider the residual structure $x_{n+1} = x_n - 2W_nT_n^{-1}\sigma(W_n^\top x_n + b_n)$ and $x_0 = x$. Denoting $\tilde{x}_{n+1} = \sigma(W_n^\top x_n + b_n)$ ($n \ge 1$) and $\tilde{x}_0 = x_0$, we can express the entire multi-layer residual structure in terms of $(\tilde{x}_0, \ldots, \tilde{x}_{N-1})$ as

$$\tilde{x}_{n+1} = \sigma\left(W_n^\top\left(\tilde{x}_0 - \sum_{k=0}^{n-1}2W_kT_k^{-1}\tilde{x}_{k+1}\right) + b_n\right), \quad y = \tilde{x}_0 - \sum_{k=0}^{N-1}2W_kT_k^{-1}\tilde{x}_{k+1} \quad (13)$$

for $n \in \{1, \ldots, N-1\}$. With this description, we can rewrite the SLL parameterization in the DEQ form of $(z, y) = (\phi(Wz + Ux + b_x), Gz + b_y)$ with $z = [\tilde{x}_0^\top, \ldots, \tilde{x}_l^\top]^\top$, $\phi(\cdot) =$

$[id(\cdot)^\top, \sigma(\cdot)^\top, \ldots, \sigma(\cdot)^\top]^{\top 3}$, $G = [I, -2W_0 T_0^{-1}, \ldots, -2W_{N-1} T_{N-1}^{-1}]$, $U = [I, 0, \ldots, 0]^\top$, and $W$ being given as the following matrix:

$$W = \begin{bmatrix} 0 & & & & \\ W_0^\top & 0 & & & \\ W_1^\top & -2W_1^\top W_0 T_0^{-1} & \ddots & & \\ \vdots & \vdots & \ddots & 0 & \\ W_{N-1}^\top & -2W_{N-1}^\top W_0 T_0^{-1} & \ldots & -2W_{N-1}^\top W_{N-2} T_{N-2}^{-1} & 0 \end{bmatrix}. \tag{14}$$

It is not surprising that the above matrix has a lower-triangular structure, since the original SLL network is feed-forward. Now it is straightforward to verify that (10) is feasible (see Appendix C) with $L = 1$ and $\Lambda$ being given by

$$\Lambda = \text{blkdiag}\left(I, 2T_0^{-1}, \ldots, 2T_{N-1}^{-1}\right). \tag{15}$$

Now we can reparameterize SLL as LBEN with preserving the 1-Lipschitz property as follows.

**Theorem 3.5.** *Given any SLL structure parameterized by $\{W_n, T_n\}_{n=0}^{N-1}$ with $W_n^\top W_n \preceq T_n$ for all $n$, the resulting input-output relation from $x$ to $y$ can be exactly recovered with the LBEN parameterization with the following choice of $(\Psi, G, U, V, S, \epsilon)$:*

$$\Psi = \Lambda^{-1} \text{ (as in Eq. (15))}, \quad G = [I, -2W_0 T_0^{-1}, \ldots, -2W_{N-1} T_{N-1}^{-1}]$$

$$U = [I, 0, \ldots, 0]^\top, \quad V^\top V = \frac{1}{2}(M + M^\top), \quad S = \frac{1}{2}M, \quad \epsilon = 0,$$

*where $M$ is given as $M = \Lambda(I - W) - \frac{1}{2}\left(G^\top G + \Lambda U U^\top \Lambda\right)$ with $W$ being defined by (14).*

A detailed proof for the above theorem will be presented in the appendix. We want to comment that one can easily verify that the expression for $V^\top V$ is given explicitly by a block-diagonal matrix with a leading zero block and other remaining blocks defined as

$$\left(V^\top V\right)_{nn} = 2T_n^{-1} - 2T_n^{-1} W_n^\top W_n T_n^{-1} \tag{16}$$

for $n \in \{1, \ldots, N-1\}$. Obviously, the resultant matrix for $V^\top V$ is positive semidefinite. We also emphasize that the above reparameterization works for both SLL and CPL, as long as one chooses $\{T_n\}$ properly. In addition, it is worth noting that one can set $\Psi = \text{diag}(\exp(d_i))$, leading to unconstrained decision variables $\{d_i\}$, which can be useful for finetuning LBEN from SLL.

**Reparameterizing AOL and Sandwich Layers as LBEN.** Notice that non-residual Lipschitz network structures such as AOL and Sandwich can also be connected to LBEN. For the experimental evaluation, we mainly utilize the SLL initialization of LBEN. We leave the discussions on the explicit connections between AOL, Sandwich, and LBEN to the appendix.

## 4 Numerical Experiments

In this section, we present some numerical experiments and insights on how to initialize LBEN with other Lipschitz architectures. Such numerical study will demonstrate and support our theory.

### 4.1 Convolutional LBEN and Implementation Details

For classification tasks, we need to embed convolutional layers into the LBEN theoretical framework. Although this is admissible in principal, there are practical aspects that need extra explanations.

**Initialization of Convolutional LBEN.** When initializing LBEN parameters $(V, S)$ from a trained SLL network, we require decomposing $\frac{1}{2}(M + M^\top)$ as constructed from Section 3 ($M$ is given by Theorem 3.5). For convolutional layers, we consider convolution with circular padding which is known to be close under sum, product, and inverse. Hence, all operations required for forming $M$ preserve the convolution structure and so $\frac{1}{2}(M + M^\top)$ is also convolution with circular padding. Furthermore, we can find $V$ by taking the matrix square root of $\frac{1}{2}(M + M^\top)$ efficiently using the block-diagonalization trick of circular convolution (see the appendix for more details).

---

[3]This just means that the first element of the activation is replaced by an identity mapping. If $\sigma$ is slope-restricted on $[0, 1]$, the resultant $\phi$ is still slope-restricted on $[0, 1]$, and hence Theorem 3.1 can still be applied.

Table 1: This table presents our results obtained from LBEN models initialized from a pre-trained Lipschitz Network (*e.g.*, MonDEQ, SLL). After initialization, the LBEN is then fine-tuned for 40 epochs with a very small learning rate. LBEN offers improved $\ell_2$-certified accuracy over Lipschitz networks counterparts. For MNIST, SLL and LBEN baselines are omitted since LBEN already achieves reasonable performance. We initialized instead from Lip-MonDEQ to showcase how our theoretical framework can provide further improvements.

| Datasets | Models | Natural Accuracy | Certified Accuracy ($\varepsilon$) | | | |
|---|---|---|---|---|---|---|
| | | | $\frac{36}{255}$ | $\frac{72}{255}$ | $\frac{108}{255}$ | $1$ |
| MNIST | (1) **Lip-MonDEQ** [31] | 0.886 | 0.886 | 0.858 | 0.826 | 0.634 |
| | **LBEN initialized from Lip-MonDEQ** (1) | 0.927 | 0.927 | 0.910 | 0.886 | 0.731 |
| CIFAR10 | (2) **SLL Network** [2] | 0.654 | 0.555 | 0.458 | 0.363 | 0.106 |
| | **LBEN** [35] | 0.451 | 0.361 | 0.277 | 0.207 | 0.043 |
| | **LBEN initialized from SLL** (2) | 0.655 | 0.562 | 0.472 | 0.380 | 0.123 |
| CIFAR100 | (3) **SLL Network** [2] | 0.398 | 0.288 | 0.207 | 0.149 | 0.038 |
| | **LBEN** [35] | 0.292 | 0.176 | 0.117 | 0.078 | 0.015 |
| | **LBEN initialized from SLL** (3) | 0.403 | 0.291 | 0.209 | 0.153 | 0.041 |

$\ell_1$ **Regularization of LBEN Sparsity Structure.** Although we endow the weight $W$ with a convolution structure, which will be preserved throughout training, this parameterization uses a relatively large kernel. Rather than using, for example, a $3 \times 3$ kernel applied across the image, our kernel parameterization is the size of the input image $n \times n$ (*i.e.*, $W$ is defined as a large kernel of size $dc_{out} \times dc_{in} \times n \times n$, where $d$ corresponds to the number of convolutions). Indeed, the inverse or square root of a convolution with a $3 \times 3$ kernel applied on an input $n \times n$ input is another convolution with a $n \times n$ kernel.

When initializing the LBEN from an existing SLL model, $W$ will be quite sparse due to the original small kernel parameterization. However, this sparsity will not necessarily be preserved after training. This is because the gradients with respect to $W$ (defined with kernel $3 \times 3$) depend on the inverse of $W$ (which has a kernel of size $n \times n$).

In order to heuristically preserve this sparsity structure and not destroy the inductive bias given by the initial architecture, we impose a small $\ell_1$-penalty to non-zero values on the border of the kernel to induce learning small-size kernels. This way, we can regularize the structure of $W$ without placing explicit constraints on the LBEN parameterization.

### 4.2  Experiments: Evaluation of Pretrained Initialization and Discussion

For the experiments, we trained MonDEQ and SLL networks that would serve as initialization for LBEN. To train a 1-Lipschitz MonDEQ, we use spectral normalization [29] on the matrix $U$ and set $m = 1$. For the SLL network (e.g., (2) in Table 1), we use 4 convolutional layers with circular padding and 2 dense layers. The convolutional and dense layers have 8 and 512 channels/features, respectively. We use the ReLU nonlinearity which is slope-restricted on $[0, 1]$. Our final LBEN network is a composition of convolutional and dense LBEN blocks, initialized from the four convolutions and the two dense layers of the pretrained Lipschitz model, respectively. To fine-tune the LBEN after initialization, we use a small learning rate of $1e-7$ with a $\ell_1$-regularization of 0.1 during 40 epochs (for more details, see the publicly available code). The results are presented in Table 1.

Note that we omit any unconstrained DEQ baselines which may have a higher clean accuracy compared to a constrained model, but very low certified robustness. The resulting Lipschitz constant of the unconstrained model is typically much larger than 1 and leads to certified accuracy near zero based on the margin argument. For instance, the work of [31] provides an evaluation of DEQ on CIFAR10 for $\varepsilon = 0.01$ achieving certified robust accuracy of roughly $10\%$ (estimate is read off graph from [31, Figure 9]). Previously, no certified robustness results for the standard perturbation $\varepsilon = 36/255$ have been reported for unconstrained DEQ.

**LBEN Initialized with Lip-MonDEQ.** First, we present a small-scale experiment on the MNIST dataset with Lip-MonDEQ and LBEN initialized from the trained Lip-MonDEQ. SLL and LBEN baselines are omitted since LBEN already achieves reasonable performance and our aim is to

showcase how our theoretical framework can provide further improvements on existing architectures. We observe that the LBEN significantly improves the natural and provable accuracy over the Lip-MonDEQ. This is mainly due to the different Lipschitz constraints used in Lip-MonDEQ and LBEN.

**LBEN initialized with SLL.** We now present some experiments on CIFAR10 and CIFAR100 datasets [20]. First, we trained an SLL network (2) with circular padding with the same hyper-parameters as in [2]. We then compare the natural and certified accuracy between an LBEN trained from scratch (random initialization) and an LBEN initialized from the SLL model. We observe that the LBEN with random initialization offers poor natural and certified accuracy compared to the SLL network for both the CIFAR10 and CIFAR100 datasets. On the other hand, when LBEN is initialized from the trained SLL weights, it successfully improves upon SLL by approximately 1% in certified accuracy on CIFAR10 and a marginal improvement on CIFAR100. Despite being marginal improvements over SLL, these results significantly improve upon the current state-of-the-art $\ell_2$-certified accuracy for DEQs.

One interpretation of these results is that the performance of neural networks is highly-dependent on a good initialization [39, 1]. Additionally, explicit feed-forward 1-Lipschitz convolutional networks have important inductive biases that have been crucial for achieving good certified robustness results on image-classification tasks [9]. In contrast, our current understanding on how to incorporate the right inductive biases for LBEN in the context of certified robustness is relative limited. Therefore, fine-tuning LBEN from 1-Lipschitz layers with good inductive biases (in our case SLL) helps LBEN achieve improved certified robustness via combining the benefits of inductive biases of feed-forward 1-Lipschitz networks and the expressive advantage of LBEN over explicit networks.

**LBEN initializations with Lipschitz constant other than $L = 1$.** Larger Lipschitz constant parameterizations for LBEN are explored in [35] on CIFAR10 ($L = 2, 3, 5, 50$), which slightly improves clean accuracy, *but decreases the empirical robustness* when compared to the 1-Lipschitz LBEN. The best certified robustness results achieved by our approach on CIFAR10 and CIFAR100 tasks are indeed achieved by choosing $L = 1$. At this moment, the understanding of how to incorporate inductive bias via enforcing convolution structures on 1-Lipschitz layers is relatively matured. Hence choosing $L = 1$ to make the Lipschitz constant consistent with these structures leads to the best certified robustness result for now. In the future, it is possible that one can improve the certified robustness of DEQ for $L > 1$ by developing new convolutional structures for LBEN.

**Remark.** *The results stated in Table 1 are not on par with the state-of-the-art certified robust accuracy obtained by the largest explicit Lipschitz feed-forward models. This is due to the following reasons: 1) DEQs are known to be computationally expensive as they require the computation of a fixed point for each input $x$. Therefore, in order to keep the computational cost for LBEN models reasonable, we trained small SLL networks. 2) There is a technical subtlety of embedding a convolutional SLL model into DEQ that requires a much larger convolutional kernel. This also currently prevents us from scaling up to larger vision tasks like TinyImageNet, but may be circumvented in the future. For applications using standard fully-connected layers, the representation memory footprint of an equivalent DEQ does not present such an issue. Despite these challenges, to the best of our knowledge, our work provides the best $\ell_2$-certified accuracy with DEQs.*

## 5 Conclusion

In this paper, we present a unified algebraic approach for deriving 1-Lipschitz DEQ structures via providing analytical solutions to an SDP condition. We show that several Lipschitz structures (*e.g.*, CPL, SLL, AOL, and MonDEQ) can be reparameterized as special cases of LBEN without changing the original Lipschitz constant specifications. We provide explicit formulas for such Lipschitz-preserved reparameterization. Finally, our experiments show that our theory opens the possibility of achieving improved certified robustness of DEQs via a combination of network reparameterization, structure-preserving regularization, and LBEN-based fine-tuning.

## Acknowledgment

A. Havens and B. Hu are generously supported by the NSF award CAREER-2048168, the AFOSR award FA9550-23-1-0732, and the IBM/IIDAI award 110662-01. A. Araujo, S. Garg, and F. Khorrami

are supported in part by the Army Research Office under grant number W911NF-21-1-0155 and by the New York University Abu Dhabi (NYUAD) Center for Artificial Intelligence and Robotics, funded by Tamkeen under the NYUAD Research Institute Award CG010.

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

# A Controlling the Lipschitz Constant in MonDEQ and G-MonDEQ

As discussed in Section 3, there are explicit relationships between the MonDEQ and G-MonDEQ parameterizations and the LBEN parameterization. Although nominally the MonDEQ only guarantees well-posedness of the DEQ, we can control the Lipschitz constants through the hyper-parameter $m > 0$, the input matrix $U$ and, in the case of G-MonDEQ, the additional parameter $\Phi$.

It was previously shown by [31] that a MonDEQ has a Lipshitz constant $L = \|U\|_2/m$. If we wish to parameterize $U$ so that MonDEQ is always 1-Lipschitz, we can choose $U = mP$ where $P$ is an orthogonal matrix. Orthogonal matrices can be parameterized by considering their sub-group of special orthogonal matrices SO(n) through the Cayley transform or matrix exponential on skew-symmetric matrices used in [41].

In the case of G-MonDEQ, a new condition is derived (Theorem 3.3) showing that the G-MonDEQ parameterizaton of (6) is $\|\Phi^{-1}U\|_2/m$-Lipschitz from $x$ to $z$. In this case, we must simultaneously control $U$ and $\Phi$ to maintain the Lipschitz bound. A similar approach can be applied to jointly parameterize $\Phi^{-1}U$. For example, to maintain a Lipschitz bound of $L$, one may practically enforce the constraints $\Phi_{ii} \geq 1/(Lm)$ and $U^\top U \preceq I$ through SLL or Cayley parameterizations [2].

# B Additional Proofs

## B.1 Proof of Theorem 3.1

*Proof.* Theorem 3.1 follows almost exactly as in [35, Theorem 2] with an added assumption of well-posedness, i.e. now it is assumed that there exists a unique solution $z = \sigma(Wz + Ux + b_z)$ for any $x$. We present the details here for completeness.

As formulated in [35], we can isolate the nonlinear parts of the DEQ through the following algebraic representation:

$$v = Wz + Ux + b_z, \quad z = \sigma(v), \quad y = Gz + b_y \tag{17}$$

Suppose that given two inputs $x$ and $x'$, the two resultant solutions for (17) are given by:

$$v = Wz + Ux + b_z, \quad z = \sigma(v), \quad y = Gz + b_y$$
$$v' = Wz' + Ux' + b_z, \quad z' = \sigma(v'), \quad y' = Gz' + b_y$$

We define $\Delta_v = v' - v$ and $\Delta_z = z' - z$. Notice that the nonlinearity $\sigma$ is assumed to be slope-restricted on $[0, 1]$. Hence the following inequality must hold for any positive definite diagonal matrix $\Lambda$:

$$\langle \Delta_v - \Delta_z, \Delta_z \rangle_\Lambda \geq 0 \tag{18}$$

where the weighted inner product is based on using $\Lambda$ as the weight. Now define $\Delta_x = x' - x$ and $\Delta_y = y' - y$, respectively. Obviously, we have $\Delta_v = W\Delta_z + U\Delta_x$ and $\Delta_y = G\Delta_z$.

Now we are ready to show the desired Lipschitz property using the matrix inequality (10). By Schur complement, we have the following equivalent matrix inequality:

$$\begin{bmatrix} 2\Lambda - \Lambda W - W^\top \Lambda - \frac{1}{L}G^\top G & -\Lambda U \\ -U^\top \Lambda & LI \end{bmatrix} \succeq 0$$

Left and right multiplying on both sides by $[\Delta_x^\top \ \Delta_z^\top]$ and $[\Delta_x^\top \ \Delta_z^\top]^\top$ respectively yields

$$2\Delta_z^\top \Lambda \Delta_z - 2\Delta_z^\top \Lambda W \Delta_z - \frac{1}{L}\Delta_z^\top G^\top G \Delta_z - 2\Delta_z^\top \Lambda U \Delta_x + L\|\Delta_x\|^2 \geq 0 \tag{19}$$

Using the relations $\Delta_v = W\Delta_z + U\Delta_x$ and $\Delta_y = G\Delta_z$, we can re-arrange the above inequality as

$$L\|\Delta_x\|^2 - \frac{1}{L}\|\Delta_y\|^2 \geq 2\Delta_z^\top \Lambda \Delta_v - 2\Delta_z \Lambda \Delta_z = 2\langle \Delta_v - \Delta_z, \Delta_z \rangle_\Lambda \geq 0. \tag{20}$$

Therefore, we must have $\|\Delta_y\| \leq L\|\Delta_x\|$, and the DEQ model (1) has to be $L$-Lipschitz. $\square$

### B.2   Proof of Theorem 3.5

*Proof.*  Given an SLL residual network, it is clear that we can define an equivalent well-posed DEQ by setting $W, U, G$ as in the theorem statement. If we substitute the parameters $(\Psi, G, U, S, \varepsilon)$ into the parameterization in (7), we are then left with

$$\frac{1}{2}(M + M^\top) = V^\top V$$

So it remains to show that $V^\top V$ is positive semidefinite, and then it is trivial to see that $V$ can be determined by decomposition.

Notice that the expression for $(M + M^\top)$ is exactly equal to the left side of the matrix inequality condition given by Eq. (10). It is simple to verify that this sufficient condition (10) is satisfied (see Section C). Therefore we have $1/2(M + M^\top) \succeq 0$. Due to a leading zero block, it is necessary that $\varepsilon = 0$. Because $1/2(M + M^\top)$ is positive semidefinite, we can always compute its decomposition (*e.g.*, via the Cholesky decomposition or matrix square-root). Therefore, $W$ can be recovered by our parameter choice $(\Psi, G, U, V, S, \epsilon)$, and the proof is complete.                    □

## C   Feasibility of SLL as LBEN

In Section 3, when discussing the connection between the SLL architecture and LBEN, it is claimed that using the weight structure:

$$W = \begin{bmatrix} 0 & & & & \\ W_0^\top & 0 & & & \\ W_1^\top & -2W_1^\top W_0 T_0^{-1} & \ddots & & \\ \vdots & \vdots & \ddots & 0 & \\ W_{N-1}^\top & -2W_{N-1}^\top W_0 T_0^{-1} & \cdots & -2W_{N-1}^\top W_{N-2} T_{N-2}^{-1} & 0 \end{bmatrix}, \quad U = \begin{bmatrix} I \\ 0 \\ \vdots \\ 0 \end{bmatrix}$$

$$G = [I, -2W_0 T_0^{-1}, \ldots, -2W_{N-1} T_{N-1}^{-1}]$$

satisfies the non-strict LBEN sufficient condition of Theorem (3.1) for $L = 1$ and $\Lambda$ given as follows:

$$\Lambda = \begin{bmatrix} I & & & \\ & 2T_0^{-1} & & \\ & & \ddots & \\ & & & 2T_{N-1}^{-1} \end{bmatrix}.$$

We can quickly prove that this is true. Plugging in $W$, $G$, $U$ and $\Lambda$ into the inequality (10), all off-diagonal block terms will cancel and we are left with the follow block-diagonal matrix:

$$\begin{bmatrix} 0 & & & \\ & 2T_0^{-1} - 2T_0^{-1} W_0^\top W_0 T_0^{-1} & & \\ & & \ddots & \\ & & & 2T_{N-1}^{-1} - 2T_{N-1}^{-1} W_{N-1}^\top W_{N-1} T_{N-1}^{-1} \end{bmatrix}. \tag{21}$$

Because it is block-diagonal, it is enough to check positive semidefiniteness of each block. Because each $T_n$ is defined in the SLL architecture [2] and satisfies $W_n^\top W_n \preceq T_n$, the following conclusion must hold:

$$2T_n^{-1} - 2T_n^{-1} W_n^\top W_n T_n^{-1} \succeq 2T_n^{-1} - 2T_n^{-1} T_n T_n^{-1} = 0.$$

Therefore, the SDP sufficient condition (10) is feasible.

# D  Further Connections between Existing Lipshitz Architectures and LBEN

## D.1  Connections between AOL and LBEN

In a similar manner, we may also reparameterize AOL as LBEN with $L = 1$. Suppose we have the following multi-layer AOL network structure

$$x_0 = x$$
$$x_{j+1} = \phi(W_j D_j x_j + b_j), \quad j \in \{0, \ldots, l-1\}$$
$$f_\theta(x) = W_l D_l x_l + b_l$$

where $D_j$ are given by (8). Then the corresponding DEQ may be defined by setting: $z = [x_1^\top, \ldots, x_l^\top]^\top$, $\sigma(z) = [\phi(z_1)^\top, \ldots, \phi(z_l)^\top]^\top$, $U = [(W_0 D_0)^\top, \ldots, 0]^\top$, $G = [0, \ldots, W_l D_l]$ and choosing $W$ as

$$W = \begin{bmatrix} 0 & & & \\ W_1 D_1 & 0 & & \\ & \ddots & \ddots & \\ & & W_{l-1} D_{l-1} & 0 \end{bmatrix}$$

It can be shown that by choosing $\Lambda = I$, the AOL network will satisfy the non-strict 1-Lipschitz LBEN condition (10). By defining $M$ and $S$ in the same way as in Theorem 3.5, we can obtain $V^\top V = \frac{1}{2}(M + M^\top)$, which can be verified to be positive semidefinite. With this choice of $V, S$ and $\Psi = I$, we have recovered $W$ by the LBEN parameterization (7).

## D.2  Connections Between Direct Sandwich Parameterizations and LBEN

Suppose we have the following multi-layer network with the Sandwich layer structure of [43] with a prescribed Lipschitz bound of $L$:

$$x_0 = \sqrt{L}x, \quad x_{n+1} = \sqrt{2}A_n^\top \Phi_n \sigma(\sqrt{2}\Phi_n^{-1} B_n x_n + b_n), \quad y = \sqrt{L}B_N x_N + b_N,$$

where $A_n A_n^\top + B_n B_n^\top = I$, $B_N^\top B_N = I$, and $\Psi_n$ is a free diagonal matrix whose entries are restricted to be non-negative. One can parameterize $[A_n \quad B_n]^\top$ as a semi-orthogonal matrix using the generalized Cayley transformation [19], and train the above network efficiently.

We can rewrite Sandwich as

$$y_0 = \sqrt{L}x, \quad y_{n+1} = \sigma(2\Phi_n^{-1} B_n A_{n-1}^\top \Phi_{n-1} y_n + b_n), \quad y = \sqrt{2L}B_N A_{N-1}^\top \Phi_{N-1} y_N + b_N.$$

Then the DEQ may be defined by setting: $z = [y_1^\top, \ldots, y_N^\top]^\top$, $\sigma(z) = [\phi(z_1)^\top, \ldots, \phi(z_N)^\top]^\top$, $U = [(\sqrt{2L}\Phi_0^{-1} B_0)^\top, \ldots, 0]^\top$, $b_x = [b_0^\top, \ldots, b_{N-1}^\top]^\top$ $G = [0, \ldots, \sqrt{2L}B_N A_{N-1}^\top \Phi_{N-1}]$, $b_y = b_N$, and choosing $W$ as

$$W = \begin{bmatrix} 0 & & & \\ 2\Phi_1^{-1} B_1 A_0^\top \Phi_0 & 0 & & \\ & \ddots & \ddots & \\ & & 2\Phi_{N-1}^{-1} B_{N-1} A_{N-2}^\top \Phi_{N-2} & 0 \end{bmatrix} \tag{22}$$

Given any Sandwich network, the input-output relationship can be exactly recovered by an LBEN parameterization with the following choice of $(\Psi, G, U, V, S, \epsilon)$:

$$\Lambda = \operatorname{diag}(\Phi_0^2, \Phi_1^2, \ldots, \Phi_{N-1}^2), \quad \Psi = \Lambda^{-1}$$
$$G = [0, \ldots, \sqrt{2L}B_N A_{N-1}^\top \Phi_{N-1}], \quad U = [(\sqrt{2L}\Phi_0^{-1} B_0)^\top, \ldots, 0]^\top$$
$$V^\top V = \frac{1}{2}(M + M^\top), \quad S = \frac{1}{2}M, \quad \epsilon = 0,$$

where $M$ is given by $M = \Lambda(I - W) - \frac{1}{2L}\left(G^\top G + \Lambda U U^\top \Lambda\right)$.

As a matter of fact, it can be shown that $V$ has the following analytical form:

$$V = \begin{bmatrix} A_0^\top \Phi_0 & B_1^\top \Phi_1 \\ & A_1^\top \Phi_1 & B_2^\top \Phi_2 \\ & & \ddots & & \ddots \\ & & & A_{N-1}^\top \Phi_{N-1} & B_N^\top \Phi_N \end{bmatrix} \tag{23}$$

# E    Additional Details of Convolutional LBEN

When it comes to image-classification tasks, one of the most useful architecture biases is the use of convolutional layers which are heavily used in all state-of-the-art SLL and AOL models, including our results. Convolutions with circular padding are straightforward to integrate into DEQ by representing them as doubly-block circulant weight matrices. In particular suppose we have an $n \times n$ image input and $K \in \mathbb{R}^{n \times n}$ a convolution kernel and define a doubly-block circulant matrix $W = \text{blkcirc}(K_0, \ldots, K_{n-1}) \in \mathbb{R}^{n^2 \times n^2}$, where $K_i = \text{circ}(K_{i,:})$. Then the 2D convolution of the vectorized input $x$ is given by the linear operation

$$W\text{vec}(x) = \text{vec}(K \star x) \tag{24}$$

where $\star$ is the convolution operation. The concept can be extended to convolutions with multiple input and output channels, where $W$ will be a block-doubly-block circulant matrix in $\mathbb{R}^{c_{out}n^2 \times c_{in}n^2}$.

This linear form has several interesting properties linked to the Fourier transform which allows efficient operations. Indeed, for initializing LBEN, we must use an expensive matrix square-root operation to compute the LBEN parameter $V$. More importantly, the operator splitting approach in [47] requires an inverse of $W$ for each step of both training and evaluation. Fortunately, there exists an efficient approach for diagonalizing this particular representation of convolutions [41] which has been used previously for convolutional MonDEQs [47]. In particular, given a doubly-block circulant weight matrix $W \in \mathbb{R}^{c_o ut n^2 \times c_i n n^2}$ with kernel $K \in \mathbb{R}^{c_{out} \times c_{in} \times n \times n}$, we can obtain its block-diagonalization as follows from [41, Corollary A.1.1]:

$$W = \left(I_{c_{out}} \otimes F_n\right) S_{c_{out}} \widehat{D} S_{c_{in}}^\top \left(I_{c_{in}} \otimes F_n^{-1}\right) \tag{25}$$

where $S_{c_{out}}$ and $S_{c_{in}}$ are permutations matrices, $F_n = U_n \otimes U_n$, and $U_n$ is a Fourier basis matrix. Here $\widehat{D}$ will be a block-diagonal matrix with $n^2$ blocks of size $c_{out} \times c_{in}$ and is given by

$$\widehat{D} = S_{c_{out}}^\top \begin{bmatrix} F_n \text{vec}\left(K_{0,0}\right) & \ldots & F_n \text{vec}\left(K_{0,c_{in}}\right) \\ \vdots & & \vdots \\ F_n \text{vec}\left(K_{c_{out},0}\right) & \ldots & F_n \text{vec}\left(K_{c_{out},c_{in}}\right) \end{bmatrix} S_{c_{in}}$$

From here, we may compute the inverse or square root of $W$, directly from the kernel $K$, more efficiently by performing these operations only on the $n$ diagonal blocks. All operations required for training LBEN using the operator splitting approach [47] will leave $W$ within the set of these doubly-block circulant matrices, so we can repeatedly apply this diagonalization trick throughout training.

