# OpenReview forum: "Exploiting Connections between Lipschitz Structures for Certifiably Robust Deep Equilibrium Models"
_NeurIPS.cc/2023/Conference — NeurIPS 2023 poster_

### Official Review · Reviewer_MZ1P · 2023-06-22

**Soundness:** 3 good
**Presentation:** 2 fair
**Contribution:** 3 good
**Rating:** 6
**Confidence:** 3

**Summary:**

Since the introduction of Deep Equilibrium Models (DEQs), many papers have been written about the certified robustness properties of DEQ models including MonDEQ and GMonDEQ. In addition, many methods have been proposed to study the certified robustness of conventional neural works, such as AOL, SLL, and Sandwich Layers. In this paper, the authors show that all methods aforementioned can be encapsulated under the LBEN framework. The authors conclude this paper by showing experiment results for the MNIST dataset.

**Strengths:**

This paper is well written and easy to read. I believe the originality and significance of this paper lies in the observation that the LBEN framework can encapsulate methods like MonDEQ and GMonDEQ. In addition, the authors also discuss the reparametrization to fit MonDEQ into the LBEN framework, which the authors correctly identified as non trivial.

**Weaknesses:**

I believe this paper would benefit from a clear discussion of the application scenarios of the proposed method. For example, when does the proposed method fail to provide satisfactory certified robustness guarantees, and when is it good? In addition, I believe non MNIST/CIFAR experiments should be run in order to demonstrate the effectiveness of the proposed method.

**Questions:**

1. The authors note in their paper that "one can potentially fine-tune the trained models from MonDEQ, SLL, and AOL by reparameterizing the networks as LBEN and initializing the LBEN training from these reparameterized models". Why would this be better than simply training an LBEN model? If there is any difference, under which circumstances would one be better than the other?

**Limitations:**

There are no negative societal impacts of this work

---

> ### Author Rebuttal · Authors · 2023-08-10
>
> Thank you for your thoughtful and thorough evaluation. We are happy to address your concerns and questions.
>
> **I believe this paper would benefit from a clear discussion of the application scenarios of the proposed method. For example, when does the proposed method fail to provide satisfactory certified robustness guarantees, and when is it good? In addition, I believe non MNIST/CIFAR experiments should be run in order to demonstrate the effectiveness of the proposed method.**
>
> This is a very interesting comment, which will add a lot of value to our draft when addressed. The scalability in training LBENs is the main open issue. Our method will provide good certified robustness results when a relatively small 1-Lipschitz structure can be used to provide useful pretrained features for the tasks at hand.
> Due to scalability issues of training DEQ, applying them to larger datasets like TinyImageNet remains very challenging. We tried our method on TinyImageNet, but it failed due to this scalability issue. Then we tried our method (LBEN initialized from a small SLL) on CIFAR-100, and our method does achieve the state-of-the-art with respect to certified robustness of any existing DEQ model on CIFAR100. By fine-tuning the LBEN initialized from SLL, we are able to see some improvements on certified robustness. Despite being marginal, these improvements are sufficient for giving us the state-of-the-art certified robustness results of DEQs on CIFAR100.
>
> |model | clean accuracy | $\epsilon = 36/255$ | $\epsilon= 72/255$ | $\epsilon = 108/255$ | $\epsilon = 1$ |
> | ---- | ------ | --- | --- | --- | --- |
> | SLL | 0.398 | 0.288 | 0.207 | 0.149 | 0.038 |
> | LBEN | 0.292 | 0.176 | 0.117 | 0.078 | 0.015|
> | LBEN with SLL init |  0.403 | 0.291 | 0.209 | 0.153 | 0.041 |
>
> Generally, explicit feed-forward convolutional models have very good inductive biases that help them extract useful features and perform well on vision tasks. Although, DEQ are technically more expressive than feed-forward networks, they lack many of these important inductive biases. In our paper we show how one may initialize a DEQ (as LBEN) with the useful features of SLL while maintaining its Lipschitz constant.
>
> Admittedly, some of the vision benchmarks are not quite state-of-the-art with respect to the best explicit feed-forward SLL models because we are using much smaller initialized models. This is due to 1) scalability issues inherent to training DEQ and 2) a technical subtlety of embedding a convolutional SLL model into a DEQ that requires a much larger convolutional kernel. This also currently prevents us from scaling up to larger vision tasks like TinyImageNet, but may be circumvented in the future. For applications using standard fully-connected layers, the representation memory footprint of an equivalent DEQ is not as much of an issue. If one wants to achieve good certified robustness on very large vision tasks ImageNet for example, explicit models may be more appropriate as of right now.
>
> We will make sure to address these subtle tradeoffs and applications in the final version of the draft.
>
> ##
>
> **The authors note in their paper that "one can potentially fine-tune the trained models from MonDEQ, SLL, and AOL by reparameterizing the networks as LBEN and initializing the LBEN training from these reparameterized models". Why would this be better than simply training an LBEN model? If there is any difference, under which circumstances would one be better than the other?**
>
> The performance of neural networks is highly dependent on good initialization. Additionally, explicit feed-forward 1-Lipschitz convolutional networks have important inductive biases that have been crucial for certifiably robust image-classification tasks.  In contrast, our current understanding on how to incorporate the right inductive biases for LBEN in the context of certified robustness is relative limited. Therefore, fine-tuning LBEN from 1-Lipschitz layers with good inductive biases (such as SLL) can help LBEN achieve good certified robustness via combining the benefits of inductive biases of feed-forward 1-Lipschitz networks and the expressive advantage of LBEN over explicit networks.

---

> > ### Comment · Reviewer_MZ1P · 2023-08-21
> >
> > Many thanks to the authors for taking time to address my questions. I have also read carefully over the feedback of other reviewers. I would leave the question of whether this paper should be accepted for the AC to decide.

---

### Official Review · Reviewer_Lz7W · 2023-07-05

**Soundness:** 3 good
**Presentation:** 4 excellent
**Contribution:** 3 good
**Rating:** 7
**Confidence:** 3

**Summary:**

The paper addresses the robustness certification of deep equilibrium models (DEQs). It proposes a novel approach that generalizes classical Lipschitz-constrained networks by presenting them as special cases of Lipschitz-bounded equilibrium networks (LBEN). The researchers' contribution is two-fold: first, they provide conditions for a DEQ to be L-Lipschitz and extend these conditions to the reparameterization of DEQs. Second, they establish a connection between SDP-based Lipschitz layers (SLL), almost orthogonal layers (AOL), sandwich layers, and LBEN. This unique approach allows for an improved certified robustness of DEQs, further enhancing the applicability and reliability of these machine learning models.

**Strengths:**

-The paper interestingly addresses the problem of Lipschitz constant certification for different types of parameterizations. This is a noteworthy contribution to the field.

-The linkage between DEQ and 1-Lipschitz neural network, considering different parameterizations, provides a novel perspective on the analysis of 1-Lipschitz neural networks. The experimental results suggest this could be an efficient approach for initializing DEQ with a Lipschitz certificate.

-The paper is very well written, self-contained, and reader-friendly. The state-of-the-art seems comprehensive and complete.


**Weaknesses:**

-The experimental section of the paper is somewhat limited and doesn't fully cover all the propositions put forth in the paper.

**Questions:**

The paper does an excellent job bridging different types of DEQs and 1-Lipschitz neural networks, which I find commendable. However, the experimental part is very limited and doesn't match the ambition of the preceding sections

-For instance, why are there no baselines with unconstrained DEQs?

-Why was a different initialization scheme used for the two experiments? SLL could have been used for both the MNIST and CIFAR datasets.

-The justification for not displaying results for AOL and sandwich layers only focuses on the robustness certification. However, AOL provides more than just Lipschitz guarantees, including quasi orthogonality. The paper would have a broader impact if it considered all different types of 1-Lipschitz parameterizations covered by the LBEN formalization in the experiments.


**Limitations:**

The authors adequately addressed the limitations

---

> ### Author Rebuttal · Authors · 2023-08-10
>
> Thank you for your thoughtful and thorough evaluation. We are happy to address your concerns and questions.
>
> **The paper does an excellent job bridging different types of DEQs and 1-Lipschitz neural networks, which I find commendable. However, the experimental part is very limited and doesn't match the ambition of the preceding sections**
>
> Thank you for your comment. Please refer to our general comment on the significance of our experimental results to see more details on why we believe our results do justify our theoretical claims.
>
> **-For instance, why are there no baselines with unconstrained DEQs?**
>
> An unconstrained DEQ will have a higher clean accuracy compared to a constrained model, however in this paper we are concerned with certified robustness. The resulting Lipschitz constant of the unconstrained model is typically much larger than $1$ and leads to certified accuracy near zero based on the margin argument.
>
> [PWK21] does an evaluation on certified robustness on MNIST and CIFAR10 but the certified accuracy obtained is very low. For CIFAR10 with $\epsilon = 0.01$ (Figure 9 in [PWK21]), the best achieved certified robust accuracy in [PWK21] is roughly 10%. No results for the standard $\epsilon=36/255$ is given.
>
> **-Why was a different initialization scheme used for the two experiments? SLL could have been used for both the MNIST and CIFAR datasets.**
>
> Sorry for the confusion. We will clarify this in our revised draft. There are a few reasons for this. In the original Lip-MonDEQ paper, MNIST is used as a benchmark, where the original SLL work focused on CIFAR benchmarks. Before our work, Lip-MonDEQ already achieved reasonable certified robustness on MNIST and we only include MNIST results to showcase how our theoretical connections between Lip-MonDEQ and LBEN can further improve a Lip-MonDEQ model. The CIFAR setting is where our method really makes a difference, since previously DEQs do not achieve good certified robustness on these tasks. Previously, SLL works well for CIFAR and hence we use SLL as initial conditions when considering CIFAR.
>
> **-The justification for not displaying results for AOL and sandwich layers only focuses on the robustness certification. However, AOL provides more than just Lipschitz guarantees, including quasi orthogonality. The paper would have a broader impact if it considered all different types of 1-Lipschitz parameterizations covered by the LBEN formalization in the experiments.**
>
> Thanks for this useful feedback. In the experiments, we have only succeeded with the SLL initializations. Investigating how to achieve good certified robustness results for LBEN initialized from AOL and Sandwich on large datasets is definitely an important future task, and we want to highlight a few challenges here. Although we do show explicitly how LBEN can be initialized from an AOL, and is indeed an equivalent DEQ network at initialization, the quasi-orthogonality property will not be preserved after further training unless additional special structure is assumed about the LBEN parameterization. In addition, AOL typically requires the use of GroupSort activation to address the gradient vanishing issue, and the current LBEN theory does not address GroupSort activations which are not sloped-restricted on [0,1]. In contrast, SLL with ReLU activations automatically address the gradient vanishing issue due to the residual network structure, and can be connected to the current LBEN theory. For sandwich layers, the current convolution forms use stride. How to incorporate consistent convolution structures for LBEN remains an open question. Although our theory does provide a clean connection between LBEN and Sandwich, how to use such a connection for vision tasks requires further study on convolutional LBEN.

---

> > ### Comment · Reviewer_Lz7W · 2023-08-18
> >
> > Thank you for addressing my query. I stand by my score and recommend accepting the paper.

---

> > > ### Author Response · Authors · 2023-08-21
> > >
> > > Thank you for considering our rebuttal. We appreciate your queries and feedback.

---

### Official Review · Reviewer_gX3G · 2023-07-06

**Soundness:** 3 good
**Presentation:** 3 good
**Contribution:** 2 fair
**Rating:** 6
**Confidence:** 2

**Summary:**

This paper demonstrates that various widely-used Lipschitz network structures, including convex potential layers (CPL), SDP-based Lipschitz layers (SLL), almost orthogonal layers (AOL), Sandwich layers, and monotone DEQs (MonDEQ), can all be reparameterized as specific cases of the Lipschitz-bounded equilibrium networks (LBEN). This reparameterization does not alter the prescribed Lipschitz constant in the original network parameterization. A notable aspect of our reparameterization technique is that it maintains the Lipschitz prescription utilized in different structures.


**Strengths:**


[Strengths]

The paper is well-structured and clearly written.

The authors offer a high-level understanding of CPL, AOL, Sandwich layers, and MonDEQ, which can all be reparameterized as specific cases of LBEN.


**Weaknesses:**

[Weaknesses]

While the theoretical results are intriguing, I believe that additional numerical results should be conducted to further validate the theorem.

The numerical results are quite preliminary and only discuss the MNIST and CIFAR10. I argue that these results do not sufficiently support the paper's theorem.

The results on CIFAR10 are not convincing. LBEN, when initialized from SLL, does not clearly outperform the SLL network.

**Questions:**

I question whether a 1-Lipschitz is necessary, as a small Lipschitz constant can significantly limit the representation ability. Why is a K-Lipschitz, where K is a limited constant, not acceptable?


**Limitations:**

Overall, while the theoretical proof is elegant, the experimental results do not adequately support the theoretical conclusions.

---

> ### Author Rebuttal · Authors · 2023-08-10
>
> Thank you for your thought and thorough evaluation. We respond to your comments as below.
>
>
> **While the theoretical results are intriguing, I believe that additional numerical results should be conducted to further validate the theorem. The numerical results are quite preliminary and only discuss the MNIST and CIFAR10. I argue that these results do not sufficiently support the paper's theorem. The results on CIFAR10 are not convincing. LBEN, when initialized from SLL, does not clearly outperform the SLL network.**
>
> Thank you for comment. We have provided additional results on CIFAR100 which are also currently state-of-the-art with respect to certified robustness of DEQ. Please refer to our general comment on the significance of our experimental results to see our explanation on why we believe our numerical results do justify our theoretical claims (our key theoretical claim is that it is crucial to maintain the prescribed Lipschitz constant when reparameterizing other Lipschitz structures as LBEN).
>
> Due to scalability issues of training DEQ, applying them to larger datasets like TinyImageNet remains very challenging. We have evaluated LBEN initialized from a small SLL network on CIFAR-100 which achieves the state-of-the-art with respect to certified robustness of any existing DEQ model. Please note there is still a significant performance gap from state-of-the-art due to scalability issues of DEQ. By fine-tuning the LBEN initialized from SLL, we are able to see some improvements on certified robustness. Despite being marginal, such improvements are enough for giving us the state-of-the-art certified robustness results for DEQs on CIFAR100.
> |model | clean accuracy | $\epsilon = 36/255$ | $\epsilon= 72/255$ | $\epsilon = 108/255$ | $\epsilon = 1$ |
> | ---- | ------ | --- | --- | --- | --- |
> | SLL | 0.398 | 0.288 | 0.207 | 0.149 | 0.038 |
> | LBEN | 0.292 | 0.176 | 0.117 | 0.078 | 0.015|
> | LBEN with SLL init |  0.403 | 0.291 | 0.209 | 0.153 | 0.041 |
>
> ##
>
> **I question whether a 1-Lipschitz is necessary, as a small Lipschitz constant can significantly limit the representation ability. Why is a K-Lipschitz, where K is a limited constant, not acceptable?**
>
> This is an interesting question and non-obvious aspect of training robust Lipschitz-constrained networks. Larger Lipschitz constants for LBEN are tried in [RWM20] on CIFAR 10 ($L=2,3,5,50$) which slightly improves the clean accuracy, but **decreases the empirical robustness** when compared to the 1-Lipschitz LBEN. We have tried to use different L to improve our certified robustness results, but we did not succeed in achieving better results with larger $L$. The best certified robustness result achieved by our approach on CIFAR10 is indeed based on choosing $L=1$. One intuitive explanation for this is that the understanding of how to incorporate inductive bias via enforcing convolution structures on 1-Lipschitz layers is relatively matured, and hence choosing $L=1$ to make the Lipschitz constant consistent with these structures leads to the best certified robustness result for now. If we use LBEN with different $L$, we will have difficulty in coming up the right features for certifiably robust classification tasks. In contrast, if we use LBEN with $L=1$, we are directly fine-tuning based on the useful features learned by 1-Lipschitz networks.
>
> - [RWM20] Lipschitz-bounded equilibrium networks

---

> > ### Comment · Reviewer_gX3G · 2023-08-18
> >
> > Thanks for your responses.
> >
> > - While I still find the experiments to be somewhat lacking in solidity, I maintain my view that this is a commendable paper. I am particularly satisfied with the high-level understanding of CPL, AOL, Sandwich layers, and MonDEQ. I raise my score to 6.

---

> > > ### Author Response · Authors · 2023-08-21
> > >
> > > Thank you for taking our rebuttal into consideration. We appreciate your updated evaluation.

---

### Official Review · Reviewer_PSUC · 2023-07-06

**Soundness:** 3 good
**Presentation:** 3 good
**Contribution:** 3 good
**Rating:** 6
**Confidence:** 3

**Summary:**

The paper studies the l2-certified robustness of DEQs from the Lipschitz bounded view. They not only proved the advantages of DEQs against other models on certifiable robustness but also show the links between other popular Lipschitz layers like convex potential layers, SDP-based Lipschitz layers, almost orthogonal layers and Lipschtiz bounded equilibrium models. Based on the relation, they can use pre-trained SLL models as initialization to help their DEQs training.

**Strengths:**

Their writing is clear. They show the relationship between other explicit Lipschitz networks and LBENs by a new reparameterizing technique. By using this technique, they use a pre-trained SLL model to help DEQ's training and obtain better certifiable results. Furthermore, I think they may show a good technique to accelerate DEQs training if their parameterization technique can extend to more model training.

**Weaknesses:**

The results for SLL are much lower than their paper's report. More than 4% lower for natural accuracy and about 10% lower for 72/255 certified accuracy compared with SLL small.

You may also compare CIFAR-100 and TinyImageNet in your empirical section.

**Questions:**

Can such fine-tuning technique help other DEQs training for natural tasks like using MDEQ[1] or MOptEqs[2]?

[1] Multiscale deep equilibrium models
[2] Optimization inspired Multi-Branch Equilibrium Models

**Limitations:**

Nothing

---

> ### Author Rebuttal · Authors · 2023-08-10
>
> Thank you for your thoughtful evaluation and comments. Now we provide detailed responses to each of your comments.
>
> ##
> **The results for SLL are much lower than their paper's report. More than 4% lower for natural accuracy and about 10% lower for 72/255 certified accuracy compared with SLL small.**
>
> We agree that this difference in performance requires more explanation. We remark that, due to scalability issues that are inherent to DEQ, we train smaller SLL networks than what is presented for the state-of-the-art in Araujo et al. The SLL network used in our paper consists of $500k$ parameters, where the``SLL Small'' architecture used in Araujo et al has $40m$ parameters, leading to the performance gap. Note that this difference is mainly due to scalability issues of DEQ, however this is still the highest certified accuracy achieved by a DEQ model. We will provide these additional architecture details in the final version.
>
> ##
> **You may also compare CIFAR-100 and TinyImageNet in your empirical section.**
>
> Due to scalability issues of training DEQ, applying them to larger datasets like TinyImageNet remains very challenging. We have evaluated LBEN initialized from a small SLL network on CIFAR-100 which achieves the state-of-the-art with respect to certified robustness of any existing DEQ model. Please note there is still a significant performance gap from state-of-the-art due to scalability issues of DEQ. By fine-tuning the LBEN initialized from SLL, we are able to see some improvements on certified robustness.
> |model | clean accuracy | $\epsilon = 36/255$ | $\epsilon= 72/255$ | $\epsilon = 108/255$ | $\epsilon = 1$ |
> | ---- | ------ | --- | --- | --- | --- |
> | SLL | 0.398 | 0.288 | 0.207 | 0.149 | 0.038 |
> | LBEN | 0.292 | 0.176 | 0.117 | 0.078 | 0.015|
> | LBEN with SLL init |  0.403 | 0.291 | 0.209 | 0.153 | 0.041 |
>
> ##
>
> **Can such fine-tuning technique help other DEQs training for natural tasks like using MDEQ[1] or MOptEqs[2]?
> [1] Multiscale deep equilibrium models [2] Optimization inspired Multi-Branch Equilibrium Models**
>
> Thank your intriguing question. In order to accommodate MDEQ[1], a modification of the LBEN theory and parameterization would be required to prove the Lipschitz property. This is because MDEQ has a special residual block DEQ structure, where the DEQ assumed in MonDEQ and LBEN are simply single layer DEQ.
>
> For MOptEqs, it is conceivable to parameterize the multiple parallel branch DEQ structure to guarantee $L$-Lipschitzness of the model and make use of their optimization approach, however this would require new theory since it is not a standard LBEN. This would be very interesting future work.

---

### Author Rebuttal · Authors · 2023-08-10

# General Response

First of all, we would like to thank each reviewer for their constructive feedback. We are glad to see that our theoretical connections on Lipschitz structures were generally well-received and there seems to be many interesting future directions. We would like to clarify some aspects of our experimental results and how they justify the claims we make in our paper.

## General comment about the significance of our experiments
Currently the understanding of the certified robustness of DEQ is very restricted, and hence our work is more exploratory in nature. However, based on the theoretical and empirical contributions, we believe it is fair to claim that our work succeeded in (i) "advancing our understanding of certified robustness of DEQ" and (ii) "improving certified robustness of DEQ on challenging tasks such as CIFAR10."

Based on our theoretical insight, we use SLL to initialize LBEN with $L=1$ (consequently the value of $\sqrt{2}L\epsilon$ is preserved) and improve the SOTA certified robust accuracy of DEQ on CIFAR10 from roughly 10% for $\epsilon=0.01\%$ (this is the result from [PWK21] which uses MonDEQ) to 64.6% for  $\epsilon=0.01\%$ and 56.2% for $\epsilon=36/255$. We were unable to obtain a remotely good certified robustness result training MonDEQ from scratch on CIFAR10. It seems that MonDEQ or G-MonDEQ have difficulty in learning useful features while maintaining the Lipschitz property and good prediction margins starting from scratch. In contrast, residual networks such as SLL are better in extracting useful features from scratch. One interpretation for our empirical results is that we use some useful features learned by SLL and then improve upon it using the expressive power of the DEQ structure. We believe our empirical results are sufficient in supporting our main theoretic idea that maintaining the value of $L$ during reparameterizing models as LBEN is important for certified robustness of DEQ.

Admittedly, some of the results are not quite state-of-the-art with respect to the best explicit feed-forward SLL models because we are using much smaller initialized models. This is due to 1) scalability issues inherent to training DEQ and 2) a technical subtlety of embedding a convolutional SLL model into a DEQ that requires a much larger convolutional kernel. This also currently prevents us from scaling up to larger vision tasks like TinyImageNet, but may be circumvented in the future. For applications using standard fully-connected layers, the representation memory footprint of an equivalent DEQ is not as much of an issue.

- [PWK21] Estimating Lipschitz constants of monotone deep equilibrium models

---

### Decision · Program_Chairs · 2023-09-21

**Decision:**

Accept (poster)

**Comment:**

This paper studies the robustness certification of deep equilibrium models. The authors show that various popular Lipschitz network structures can all be reparameterized as special cases of the Lipschitz-bounded equilibrium networks. The results are interesting and clearly presented.